# Detection of Porcine Circovirus Type 3 in Serum, Semen, Oral Fluid, and Preputial Fluid Samples of Boars

**DOI:** 10.3390/vetsci10120689

**Published:** 2023-12-04

**Authors:** Song Qi, Qiyun He, Zhewei Zhang, Huanchun Chen, Luis Giménez-Lirola, Fangyan Yuan, Weicheng Bei

**Affiliations:** 1National Key Laboratory of Agricultural Microbiology, College of Veterinary Medicine, Huazhong Agricultural University, Wuhan 430000, China; qisong@webmail.hzau.edu.cn (S.Q.); zhangzw@webmail.hzau.edu.cn (Z.Z.); chenhch@mail.hzau.edu.cn (H.C.); 2The Cooperative Innovation Center for Sustainable Pig Production, Huazhong Agricultural University, Wuhan 430000, China; 3Hubei Hongshan Laboratory, Wuhan 430000, China; 4Veterinary Diagnostic and Production Animal Medicine, College of Veterinary Medicine, Iowa State University, Ames, IA 50011, USA; 5Hubei Key Laboratory of Animal Embryo and Molecular Breeding, Institute of Animal Husbandry and Veterinary Sciences, Hubei Academy of Agricultural Sciences, Wuhan 430000, China

**Keywords:** PCV3, serum, semen, oral fluid, preputial fluid

## Abstract

**Simple Summary:**

Porcine circovirus type 3 (PCV3), a virus with various detrimental effects on pigs including respiratory disease, digestive disorders, congenital tremors, rectal prolapse, reproductive problems, and multisystemic inflammation, is widespread in pig populations worldwide. This research aims to examine the transmission, risk factors, mode of transmission, and epidemiology of PCV3 originating from boars to pig farms. The investigation involves analyzing fresh semen, as well as matched serum, oral fluid, and preputial hydrocele samples collected from 28 boars on a large-scale pig farm located in Guangxi, China. The study reveals that the PCV3 DNA is commonly found in oral fluid (64.28%) and preputial fluid (46.4%), but rarely detected in serum (3.57%), with no detection in semen. This is the first documented case of PCV3 detection in preputial fluid from boars. The results indicate that PCV3 spreads among boars on pig farms and exhibits epidemic characteristics. The paragraph provides insights for monitoring, preventing, and controlling PCV3 on pig farms. It highlights the importance of manually removing preputial fluids before semen collection to reduce the risk of contamination in both the semen and the collection vessel. Additionally, it offers other recommendations for preventing and controlling PCV3 in pig farms.

**Abstract:**

Porcine circovirus type 3 (PCV3) is commonly associated with clinical symptoms such as porcine dermatitis and nephropathy syndrome (PDNS)-like lesions, respiratory signs, and reproductive disorders. This study aimed to investigate the epidemiology of PCV3 in a boar stud. The objectives were to detect PCV3 in semen, as well as matched serum, oral fluid, and preputial fluid samples from adult boars using quantitative polymerase chain reaction (qPCR), analyze PCV3-IgG antibody data, and genetically characterize a positive sample. A total of 112 samples from 28 boars were collected from a large-scale pig farm in Guangxi, China. The qPCR results showed that the PCV3 DNA was not detected in semen, with a positive rate of 0% (0/28), while it was detected in serum (3.57%—1/28), oral fluid (64.28%—18/28), and preputial fluid (46.4%—13/28). The seropositivity rate of PCV3-IgG in serum was 82.14% (23/28) according to the indirect enzyme-linked immunosorbent serologic assay (ELISA) results. Phylogenetic analysis revealed that one of the PCV3 isolates belonged to the PCV3c clades. This is the first report of PCV3 detection in preputial fluid from boars. The results suggest that PCV3 is transmitted among boars on pig farms and exhibits epidemic characteristics.

## 1. Introduction

Porcine circoviruses, belonging to the family Circoviridae within the genus Circovirus, are characterized as small, non-enveloped viruses. These viruses possess single-stranded circular DNA genomes, which typically range from approximately 1.7 to 2.0 kilobases in size [1]. Until now, four types of porcine circovirus have been identified: porcine circovirus type 1 (PCV1), porcine circovirus type 2 (PCV2), porcine circovirus type 3 (PCV3), and porcine circovirus type 4 (PCV4). PCV1 is considered nonpathogenic in pigs and is first detected as a cell-culture contaminant in porcine kidney cells (PK-15) [2]. PCV2, the primary etiological agent of porcine circovirus-associated disease (PCVAD), is one of the most important pathogens that affect the swine industry, which can cause typical PDNS, postweaning multisystemic wasting syndrome (PMWS), and reproductive disorders, leading to severe economic losses worldwide [3,4,5].

Meanwhile, PCV3 was first discovered in the United States by next-generation sequencing (NGS) methods in swine with respiratory and neurological signs, cardiac, multisystemic inflammation, reproductive failure, and a symptom similar to PDNS [6,7,8]. Since then, PCV3 has been detected in many countries, including China, South Korea, Thailand, Denmark, and Italy. In the years following its discovery, PCV3 was extensively observed in various regions of China, specifically in Fujian, Hebei, Henan, Hunan, Jiangsu, Jiangxi, Liaoning, Guangxi, Shenyang, and Zhejiang provinces, through the examination of stillborn pigs or serum samples [9,10]. Currently, there exists limited research regarding the pathogenic nature of PCV3. However, numerous studies have been conducted on the identification of the PCV3 virus in preserved remains as well as its association with reproductive complications, including stillbirth and abortion [11]. A total of 1760 clinical tissue samples were collected from pigs in Henan Province, spanning 18 distinct regions, during the period between October 2018 and September 2019. These samples were subjected to rigorous screening to identify the presence of PCV2 and PCV3. The result indicated that PCV2 and PCV3 are prevalent throughout the year, and the prevalence of PCV2 is extremely high, while the prevalence of PCV3 is low but on the rise [12]. Unexpectedly, PCV3 was also widely identified in healthy pigs [13]. PCV4, a novel circovirus discovered in April 2019 in Hunan Province in a pig with PDNS, respiratory, and intestinal diseases, shows a high genomic identity to mink circovirus (66.9%) and has identities of 43.2–51.5% with other PCV genomes [14,15].

PCV3 can be detected in pigs of different ages and causes clinical symptoms similar to those caused by PCV2. PCV3 infection can cause several clinical-pathological outcomes, such as respiratory disease, digestive disorders, congenital tremors, rectal prolapse, reproductive problems, and multisystemic inflammation, which are no less harmful to the pig industry than PCV2 [16]. However, the transmission characteristics of PCV3 in infected pigs are unclear. It has been reported that PCV3 can be detected in boar semen [17,18,19], indicating the possibility of epidemic and transmission of PCV3 through boars on pig farms, but the number of samples is limited. This study aimed to further clarify the transmission risk of PCV3 from boars on pig farms by testing serum, semen, oral fluid, and preputial fluid, as well as to provide some reference for monitoring, prevention, and control of PCV3 on pig farms.

## 2. Materials and Methods

### 2.1. Reagents

The PCV3 fluorescence quantitative polymerase chain reaction (qPCR) kit and the PCV3-IgG indirect ELISA kit, meticulously developed by Luis Giménez Lirola, were obtained from Biostone Animal Health LLC (Southlake, TX, USA) and their sensitivity and specificity were thoroughly evaluated [20]. A DNA nucleic acid extraction kit (magnetic bead method) was purchased from Guangzhou Super Pure Biotechnology Co., Ltd. (Guangzhou, China). The protease K was included in the DNA nucleic acid extraction kit.

### 2.2. Animal Sources

Animal sources were obtained from a total of 28 boars, ranging in age from 18 to 35 months, which were sourced from a pig farm located in Guangxi Province and were introduced to the pig farm once they had reached at least 1 year of age. These boars were housed in pig pens with a column structure, comprising iron bars and mechanical ventilation. It is worth noting that the stillbirth rate, which ranged from 8% to 10%, as well as the mummy rate, which ranged from 2% to 4%, were observed to be higher than the previously recorded levels across all the parities since the year 2021. Utilizing the PCV3 quantitative PCR kit, variable positive rates were detected in colostrum, stillbirth, and mummified sows at different parities. However, it is important to highlight those other viruses associated with reproductive disorders, including porcine reproductive and respiratory syndrome virus (PRRSV), pseudorabies virus (PRV), porcine parvovirus virus (PPV), classical swine fever virus (CSFV), and PCV2, were not identified during the regular quarterly monitoring.

### 2.3. Samples

#### 2.3.1. Oral Fluid Collection

We used a special saliva sampling bite rope (sterilized package, purchased from Shanghai Chuanghong Biotechnology Co., Ltd., Shanghai Municipality, China), and feeding and water were limited for at least 2 h before sample collection. We opened the end of the cord to make it more absorbent and then hung the other end of the cord on the steel rail of the pen, keeping the end of the cord at the height of the boar’s shoulder. We guided the boar to chew the cord for at least 10 min. After the bite rope was fully saturated with boar saliva, the bit end was covered with a sterilized sealing bag. The oral fluid was then extracted under force and divided into 15 mL sterilized centrifugal tubes. The oral fluid samples, all of which contained no less than 5 mL, were then centrifuged at 4 °C for 10 min at 1000× *g* using a refrigerated centrifuge. Next, the supernatant was divided into 5 mL sterilizing centrifuge tubes and stored in a refrigerator at 4 °C for next testing.

#### 2.3.2. Preputial Fluid and Semen Collection

The prepuce, commonly referred to as the external penile sheath, is the encasing fold of skin that envelops the distal part of the penis when it is in a retracted position. In the case of pigs, the foreskin protrudes dorsally, creating a preputial diverticulum, which serves as a reservoir for preputial fluid. This fluid primarily consists of waste matter such as cellular debris and urine, giving it a distinctive malodorous scent. To ensure proper hygiene and cleanliness, a 0.25% Virkon solution was utilized to clean and disinfect both the semen-collecting column and the artificial livestock table. Prior to collecting semen, the staff members were provided with instructions to wipe the boar’s abdomen and urethra using a towel soaked in the aforementioned Virkon solution. Following this step, both areas were rinsed with sterilized physiological saline and dried with sterilized paper towels. For the collection of the extruded preputial fluid, a pre-prepared 100 mL sterilized beaker was employed. The initial extrusion of the preputial fluid was not collected; instead, the subsequent extrusions, totalling no less than 10 mL, were collected. These samples were divided into 15 mL centrifuge tubes and stored in a refrigerator at a temperature of 4 °C for future use. To ensure the integrity of the data obtained from subsequent tests, we took several steps to collect semen, following the collection of the preputial fluid. Firstly, the staff members trimmed the hair around the boars’ preputial opening and removed as much preputial fluid as possible. Next, they washed their hands with a 0.25% Virkon solution to minimize contamination. They then proceeded to clean and disinfect the boar’s urethral orifice and abdomen using the same solution, along with sodium chloride physiological solution and sterilized paper towels to clean the preputial opening and the surrounding area. To maintain cleanliness, double gloving was utilized, with the outer glove discarded after preparing the boar. This ensured a clean gloved hand for grasping the penis. During the semen collection, the penis was held horizontally to minimize contamination of the semen and the semen collection vessel. To safeguard against potential contamination, a minimum of 5 mL of semen was collected from the final third of the section by using a pre-prepared 100 mL sterilized beaker. This approach deviates from the conventional semen collection process for artificial insemination (AI), as it allows us to safeguard against the possibility of any contamination in the acquired semen. The collected semen samples underwent the process of freezing-thawing three times, utilizing liquid nitrogen and a water bath at a temperature of 70 °C. Finally, these samples were stored in a refrigerator at 4 °C for further testing.

#### 2.3.3. Blood Collection

Once the boars were restrained, the blood was collected from the jugular vein using a blood collector. The obtained blood was then allowed to stratify naturally, and the resulting serum was carefully poured into sterilized centrifuge tubes with a volume of 5 mL. Subsequently, the serum underwent additional centrifugation. Finally, the clear serum was divided equally among sterilized centrifuge tubes with a capacity of 5 mL each and stored in a refrigerator set at a temperature of 4 °C for future testing.

### 2.4. DNA Extraction and qPCR

The study commenced by collecting serum, semen, oral fluid, and preputial fluid samples at an early stage. To prepare the samples, we introduced 20 μL of protease K to each 200 μL sample, thoroughly mixing them and allowing them to rest for 60 s. Subsequently, the prepared samples were added to the designated well of an automatic DNA nucleic acid extraction kit and processed using an automated nucleic acid extraction instrument. Following the extraction steps outlined in the kit instructions, we successfully obtained the nucleic acid. The extracted material was stored at a temperature of −20 °C in a refrigerator for future utilization. Next, we assembled the reaction system for the PCV3 fluorescent quantitative PCR kit based on the provided guidelines. The reaction system consisted of 19 μL of PCR reaction solution, 1 μL of enzyme mixture, and 5 μL of PCV3 sample DNA template. This resulted in a total reaction volume of 25 μL for qPCR amplification. The cycling conditions entailed an initial denaturation step at 95 °C for 3 min, followed by 40 cycles of denaturation at 95 °C for 15 s and annealing at 55 °C for 30 s. A cut-off for positive samples was established at cycle threshold (CT) values lower than 40.

### 2.5. PCV3 Genome Sequencing and Bioinformatics Analyses

In order to obtain a comprehensive genome of PCV3, a previously reported PCR primer pair designed for sequencing was employed [21]. The PCR procedure was performed in a final volume of 50 μL, comprising of 4 μL DNA, 1 μL of each 20 μM primer, 2 μL (2.5 mmol/L) dNTPs, 10 μL PCR buffer, 1 μL DNA polymerase from TransGen Biotech located in Beijing, China, and ddH2O adjusted to 50 μL. The PCR profile conditions consisted of an initial denaturation at 95 °C for 2 min, followed by 40 cycles at 95 °C for 20 s, 50 °C for 20 s, and 72 °C for 40 s. A final extension step was performed at 72 °C for 5 min. Each fragment was amplified and purified using a gel extraction kit from Bioer Technology in Hangzhou, China, and subsequently cloned into a pEASY-Blunt vector from TransGen Biotech in Beijing, China. One positive clone was selected for sequencing, which was conducted by Tsingke Biotech Co. in Wuhan, China.

To conduct the phylogenetic analysis, we examined a single complete PCV3 genome sequence obtained from our research. To provide a comparison and establish a reference sequence, we utilized various PCV3 strains that were isolated from different regions within China and other countries. The selection of these strains was based on the availability of data from the NCBI nucleotide database. To align the multiple sequences, we employed the Clustal W program within the Lasergene 7.0 software. Using the MEGA 7.0 software, we reconstructed a phylogenetic analysis of the complete PCV3 genome. This analysis utilized maximum likelihood (ML), and we performed 1000 bootstrap replicates to ensure statistical robustness.

### 2.6. PCV3-IgG Antibody Detection in Serum

In accordance with the instructions provided by the manufacturer, a PCV3-IgG indirect ELISA kit which is suitable for the detection of serum samples according to the manufacturer’s instructions, was employed for the purposes of this study. In order to minimize any potential non-specific antigen-antibody interactions, all samples were subjected to a heat inactivation step at a temperature of 56 °C for 30 min. The optical density (OD) value was determined at a wavelength of 450 nm using a microplate reader. Subsequently, the PP value of each sample was calculated using the formula: (sample OD value divided by the mean OD value of the positive control) multiplied by 100%. It is worth noting that a PP value below 40% is indicative of a negative result, while a PP value exceeding 40% would suggest a positive outcome.

### 2.7. Statistical Analysis

Fisher’s exact test and the Kruskal–Wallis H test were used to compare the difference in PCV3 positive detection rate and viral load in serum, semen, oral fluid, and preputial fluid, respectively. *p* < 0.05 was considered statistically significant, and the reported mean was mean ± standard deviation.

## 3. Results

### 3.1. Detection of PCV3 in Serum, Semen, Oral Fluid, and Preputial Fluid of Boars

Table 1 presents the detailed test results. The positive detection rate and CT values of PCV3 in various bodily fluids of boars, including serum, semen, oral fluid, and preputial fluid, exhibited significant differences. Specifically, the detection rate of PCV3 in semen was 0% (0 out of 28 samples), in serum it was 3.57% (1 out of 28 samples), in the oral fluid it was 64.28% (18 out of 28 samples), and in the preputial fluid, it was 46.4% (13 out of 28 samples). The highest detection rate was observed in oral fluid, followed by preputial fluid. The corresponding data can be found in Table 2.

The study examined a total of 28 boars to determine the rate of PCV3 detection in different samples. Specifically, the rate of PCV3 detection solely in oral fluid, excluding serum, semen, or preputial fluid samples, was found to be 32.1%. Additionally, the rate of PCV3 detection solely in preputial fluid, excluding the other three types of samples, was determined to be 14.3%. Furthermore, the positive detection rate in both preputial fluid and oral fluid was observed to be 28.6%. Moreover, the positive detection rate in all three types of samples (serum, oral fluid, and preputial fluid) was found to be 3.6%. These findings have been summarized in Table 3. However, in 6 boars (21.4%), PCV3 was not detected in any of the samples, the data can be found in Table 1.

Significant variations were detected in the viral load of PCV3 in different kinds of samples (serum, semen, oral fluid, and preputial fluid), by evaluating the CT value of qPCR. The CT value, which can serve as an indicator of viral load, demonstrated a negative correlation, meaning that higher CT values corresponded to lower viral loads. The mean CT value of positive oral fluid samples was 34.63, with a maximum value of 38.29 and a minimum value of 28.9. In the case of positive preputial fluid samples, the mean CT value was 32.41, the maximum value was 37.49, and the minimum value was 28.1. Significantly, in the case of boar number 6, which was the sole boar where PCV3 was observed in the serum, the CT value of the serum was measured at 35.5, whereas the CT values for both oral fluid and preputial fluid were determined to be 28.9 and 31.28, respectively. It is important to mention that no positive PCV3 was detected in the semen.

### 3.2. Phylogenetic Analysis Based on PCV3 Complete Genome Sequences

In order to better understand the phylogenetic relationship and evolution of PCV3, an ML phylogenetic tree was constructed using the whole genome sequence of PCV3 to reconstruct the phylogeny. A total of 27 complete genome sequences available in the NCBI database were compared with the complete genome sequences (OR636679) obtained from preputial fluid samples in this study. Our study was consistent with previous research showing that PCV3 strains are divided into two distinct clades, PCV3a, PCV3b, and PCV3c (Fu et al., 2018). The PCV3 strain identified in this study belonged to the PCV3c clades. The phylogenetic tree showed this PCV3 strain was closely related to the USA and South China isolates (Figure 1).

### 3.3. Comparison of the Detection of PCV3-IgG in Serum

Table 1 presents the PCV3-IgG results. In the cohort of 28 boars that were examined, a notable finding was that 25 of them, constituting approximately 89.28% of the sample, exhibited a positive presence of PCV3-IgG antibody. Boar No. 13 which specifically detected the presence of PCV3 only in the preputial fluid, demonstrated the highest PP value of 158.8%. Among the 3 boars that yielded negative results for PCV3-IgG antibody, it was observed that 2 of them (No. 4 and No. 22) exhibited detectable shedding of PCV3 in their oral fluid or preputial fluid. Conversely, only 1 boar (No. 20) displayed no traces of PCV3 across all four sample types and was also negative for PCV3-IgG antibody. Among the four kinds of samples analyzed, it was found that a total of 6 boars, out of which five were confirmed to be positive for PCV3 antibodies, did not exhibit any detection of PCV3. The mean value of PP was calculated to be 97.45%, with a standard deviation of 35.46% and a dispersion of 0.36.

## 4. Discussion

This study employed qPCR to conduct a systematic analysis of viremic levels in various bodily fluids of boars, namely serum, semen, oral fluid, and preputial fluid. Furthermore, in this study, the assessment of the PCV3-IgG levels in the serum was carried out using a PCV3-IgG indirect ELISA kit, which was developed by Luis Giménez Lirola. The sensitivity and specificity of the kit have already been examined using caesarean-derived and colostrum-deprived (CD/CD) pigs [22].

Currently, it is well-established that boar semen can serve as a means of transmitting numerous viruses, including PCV2, PPV, PRRSV, PRV, and CSFV, among others. These viruses impose significant reproductive challenges on pig farms, leading to increased rates of estrus return, mummified fetuses, stillbirth, abortion, and substantial economic losses [23,24,25,26]. Nevertheless, the transmission of PCV3 through boar semen remains inconclusive, with only a few studies suggesting the presence of PCV3 in boar semen [12,17,18]. This scarcity of information greatly hinders our comprehension of the significance of semen in the horizontal transmission of PCV3. Moreover, these studies suffer from limited sample sizes, thus failing to comprehensively demonstrate the potential transmission of PCV3 through AI. Our study pioneers the investigation and analysis of fresh semen, accompanied by matched serum, oral fluid, and preputial fluid, with the objective of enhancing our comprehension of PCV3 transmission within pig farms. Furthermore, this study describes the first known identification of PCV3 from preputial fluid samples of boars.

It is of significance to mention that out of the total of 28 boars that were investigated, PCV3 was detected solely in the serum of a single boar. Interestingly, this particular boar exhibited the presence of PCV3 in both the oral fluid and preputial fluid, while no trace of the virus was observed in the semen. Similarly, the other 27 boars did not show PCV3 presence in their semen, which contrasts with the findings of previous research studies that semen used for AI was not likely to be important in viral transmission [18,19]. These findings suggest that PCV3 may have limited ability to penetrate the blood-testosterone barrier and transmit through semen. However, in the examined pig farm, sows displayed elevated rates of stillbirths, mummified fetuses, and estrus return and a higher incidence of PCV3 detection was observed in stillbirths and mummified fetuses, regardless of parity. Because of the significantly high detection rate of PCV3 at 46.4% in preputial fluid, we can suspect that PCV3 has the potential for virulence and horizontal transmission through boar semen by preputial fluid’ contamination. In the past few years, there have been limited studies that have discussed the identification of disease-causing agents in the preputial fluid of boars. Examples include the detection of atypical porcine pestivirus (APPV) [27] and aerobic mesophiles [28]. However, due to the scarcity of available information, our comprehension of the significance of semen in the horizontal transmission of certain diseases remains restricted. Numerous datasets indicate the necessity of manually removing preputial fluids before exposing the penis for semen collection, in order to reduce the risk of preputial fluid contamination in both the semen and the vessel used for collection [29]. This study represents the primary effort to observe the presence of PCV3 in preputial fluid and raises the possibility that semen could be probably contaminated with preputial fluid during the collection process, leading to the potential transmission of PCV3 through artificial insemination. Thus, it is crucial to conduct further investigation to closely monitor the presence of PCV3 in the semen of individual boars. This is especially important in cases where PCV3 is detectable in the serum, as the extent of contamination by preputial fluid in this field remains unknown. An additional factor contributing to the notably low rate of PCV3 detection in semen may be attributed to the limited efficacy of nucleic acid extraction kits. It has been well-documented that sperm cells possess distinctive attributes, including a compact nuclear chromatin structure in mature sperm, which presents a heightened difficulty in extracting DNA from semen when compared to blood and tissue cells [30]. Alternatively, PCV3 might exhibit intermittent shedding in semen, similar to PRRSV in semen and oropharyngeal samples [31,32], which necessitates further investigation. Notably, the diminished rate of PCV3 detection in serum could potentially be attributed to the characteristics of the sample. Several studies have indicated that PCV3 has a greater propensity to replicate and propagate within mononuclear macrophages [8]. This finding implies the importance of employing whole blood as the preferred method for detecting PCV3 in future studies. Similarly, the presence of numerous shed mononuclear macrophages in the preputial fluid can be considered a contributing factor to the elevated detection rate of PCV3 in the fluid of the prepuce.

Consequently, these findings provide valuable insights for enhancing biosafety precautions and standard operating procedures during the semen collection process in pig farms. Furthermore, PCV3 was exclusively detected in preputial fluid, with no presence in oral fluid or serum, in 14.3% of the four types of samples. This highlights the potential of using preputial fluid as an effective monitoring sample for PCV3 in boars, particularly as a preliminary step prior to performing conventional semen collection procedures. 

This study effectively monitored the occurrence of PCV3 in boars by analyzing oral fluid samples. A total of 18 samples tested positive, indicating the highest positivity rate of 64.28% compared to the other three kinds of samples. The possible reason for this result is that the boar stud’s particular housing arrangement facilitates iron bars, which increases the contact between the mouth and nose with each other. One potential explanation for this outcome is that the configuration of the boars’ enclosure, such as the presence of iron bars, enhances the proximity and interaction between their mouths and noses. Moreover, the transmission of contaminants via semen collection stations, the airborne dispersion of pathogens, and the dissemination of dust particles are also crucial contributing factors. In order to minimize the risk of PCV3 oral and nasal transmission, it is important for us to take into account the aforementioned factors during the actual production management process. To gain a deeper understanding of the horizontal transmission of PCV3 in boars, it is recommended to conduct ongoing follow-up studies on recently introduced boars.

However, the CT values obtained from PCV3 qPCR analysis of these oral fluid samples exhibited a relatively diverse distribution. The CT value ranged from a minimum of 28.9 to a maximum of 37.65, with an average value of 34.63. On the other hand, the preputial fluid sample set comprised 13 samples that tested positive, thus indicating a positivity rate of 46.43%, and the CT value ranged from a minimum of 28.1 to a maximum of 37.49, with an average value of 32.41. These findings suggest that oral fluid had a higher rate of detection compared to preputial fluid.

In this study, the full-length genome sequence of this PCV3 isolate was determined using metagenomics sequencing. Phylogenetic analysis has been widely used in the characterization of PCV3 sequences from clinical samples in multiple countries [33]. This isolate described in this study shared 98.47–99.57% nucleotide similarities with the available PCV3 reference strains from the NCBI GenBank at the complete genome sequences, this implies that PCV3 has high genetic stability, which is concordant with findings from previous studies [34]. Furthermore, the PCV3 isolate identified in this study was classified as subtype 3c, which aligns with the findings of another investigation that analyzed serum samples obtained from 2568 clinically healthy pigs across 36 large-scale pig farms in 17 provinces (Hunan, Xinjiang, Jilin, Zhejiang, Jiangsu, Jiangxi, Guangxi, Hebei, Shandong, Shanxi, Anhui, Yunnan, Hubei, Inner Mongolia, Henan, Sichuan, and Guizhou) between 2019 and 2020. The latter study reported that 86.96% (20 out of 23) of the PCV3 isolates were identified as PCV3c strains [35], indicating the existence of diverse PCV3 strains circulating in China.

There is a lack of available information regarding the immune response to PCV3. However, field studies have indicated a high seroprevalence of PCV3 in grower-finisher pigs, ranging from 22% to 80%. Furthermore, the seroprevalence in sows can reach up to 96%, suggesting a significant level of virus circulation [36]. Gun Temeeyasen et al., conducted a study that focused on the investigation of PCV3’s ability to elicit an IgG response in pigs infected with CD/CD pig. The research findings revealed that PCV3 has the potential to stimulate an early IgG response within 7 days post-infection, which is sustained for 42 days post-infection [22]. Besides, Zhang et al. [37] conducted a study that indicates that pigs exhibiting high levels of viremia also display elevated levels of IgG, whereas pigs with low levels of viremia demonstrate minimal to no detectable IgG. They also referenced the PCV3-IgG indirect ELISA methods developed by Luis Giménez Lirola [20,22]. These results provide evidence for a potential correlation between the levels of PCV3 viremia and the presence of PCV3-IgG. The findings of this research indicate that there is a remarkably high positive rate of PCV3-IgG in serum, reaching 89.28%. This suggests a significant prevalence of infection. It is noteworthy to highlight that among the three boars that yielded negative results for PCV3-IgG antibodies, only one of them did not exhibit any presence of PCV3 in the four types of collected samples. Additionally, it is intriguing to observe that out of the six boars that did not detect PCV3 in any of the four types of samples, five of them actually tested positive for PCV3 antibodies. The provided data establishes that there was no observed correlation between the presence of PCV3-IgG antibodies in the serum and the presence of PCV3 antigens in either the oral fluid or preputial fluid. 

A potential limitation of this study is the relatively small sample size, consisting of only 28 boars. As a result, the generalizability of the findings to the broader population of boars in different pig farms may not be entirely convincing. It is especially noteworthy that our research failed to investigate the correlation between boar age and the presence of PCV3-IgG antibodies as well as PCV3 shedding in serum, oral fluid, and preputial fluid. The study conducted by Matthias et al. [19] yielded compelling evidence of a discernible correlation between age and the detection rate of PCV3 in serum. Nevertheless, the epidemiological investigation of PCV3 through the examination of boar serum and matched samples such as oral fluid, preputial fluid, and semen still holds significant value in providing valuable guidance.

Overall, this study has made a significant discovery by demonstrating, for the first time, the presence of PCV3 in preputial fluid among boars on this particular pig farm. This finding offers crucial insights into the ecology and epidemiology and identifies a potential mode of transmission of PCV3 in boar studs, specifically to gilts and sows, and should be considered in the development and implementation of management and biosecurity measures to prevent and/or control clinical signs attributed to PCV3 infection. Furthermore, by incorporating data from antigen and antibody detection of PCV3 in serum, as well as antigen detection in oral fluid and semen, this study offers valuable insights for the comprehensive prevention and control of PCV3 in this field. However, it is worth noting that the age and the historical infection of the boars were not analyzed in this study, thus necessitating further research to determine the applicability of these results to other boar studs.

## 5. Conclusions

We concluded that PCV3 was less likely to be transmitted through boar semen and was not detected even in boar semen with viremic. However, the proportion of PCV3 detected in the preputial fluid of boars was as high as 46.4%, which provided another possible reference for the contamination of preputial fluid during semen collection and subsequent transmission of PCV3 through semen. The positive rate of PCV3 in oral fluid was up to 64.28%, indicating the transmission route of PCV3 through direct oral–nasal contact. The results of the phylogenetic analysis revealed that PCV3 exhibited a discernible division into three distinct clades. Importantly, the strains that were isolated in our study were found to be distributed among the PCV3c clades. The positive rate of PCV3-IgG was up to 89.28%, but the negative or positive of the antibody was not associated with viremic. Therefore, further investigation is needed to understand the relationship between PCV3-IgG and historical infection.

## Figures and Tables

**Figure 1 vetsci-10-00689-f001:**
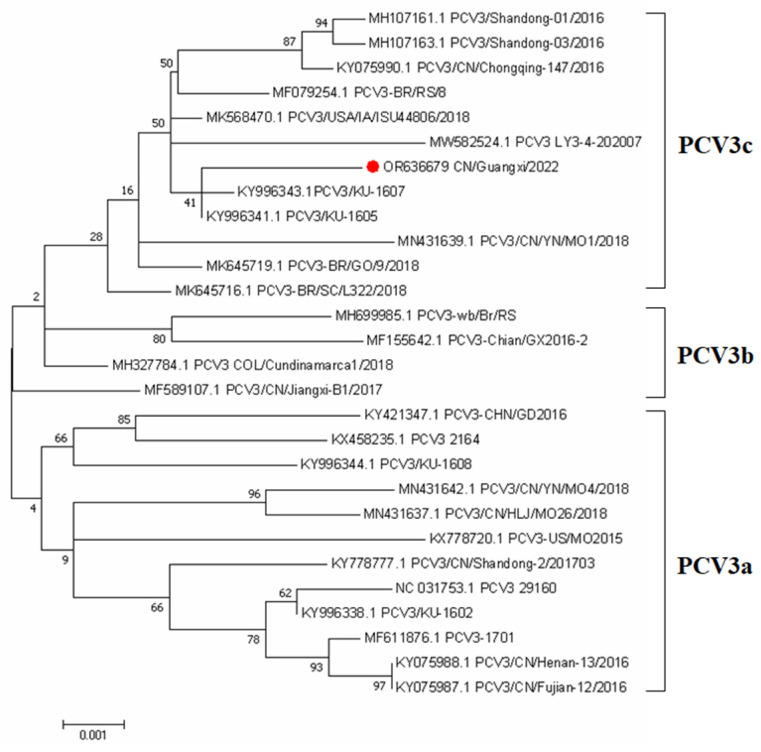
Phylogenetic analysis of the PCV3 based on complete coding sequences. The tree was constructed using MEGA version 7.0 with 1000 bootstrap replicates and a p-distance model. The PCV3 strain collected from Preputial fluid is indicated with a red roundness.

**Table 1 vetsci-10-00689-t001:** CT values of PCV3 in different samples.

Boar No.	qPCR Results (Ct)				ELISA PP (%)	Boar No.	Serum	Semen	Oral Fluid	Preputial Fluid	ELISA PP (%)
	Serum	Semen	Oral Fluid	Preputial Fluid							
1	0	0	34.36	33.04	52.84	15	0	0	37.65	31.97	142.91
2	0	0	0	37.49	88.22	16	0	0	0	0	124.62
3	0	0	33.03	0	119.35	17	0	0	35.48	28.1	61.89
4	0	0	33.68	35.86	31.96	18	0	0	0	31.71	65.77
5	0	0	0	0	100.60	19	0	0	36.45	0	74.27
6	35.5	0	28.9	31.28	83.00	20	0	0	0	0	21.89
7	0	0	35.77	0	95.61	21	0	0	34.11	33.38	93.49
8	0	0	30.82	30.38	150.67	22	0	0	33.94	0	22.73
9	0	0	0	0	79.26	23	0	0	0	0	76.30
10	0	0	38.29	0	76.86	24	0	0	36.74	31.5	98.00
11	0	0	33.83	30.71	80.00	25	0	0	36.81	0	102.00
12	0	0	33.81	0	85.08	26	0	0	0	30.52	123.81
13	0	0	0	35.36	158.80	27	0	0	35.54	0	88.90
14	0	0	34.14	0	141.71	28	0	0	0	0	72.55

Samples with CT value < 37.0 for either species were considered positive, while CT value > 40 is negative, and 37 < CT value < 40 with typical amplification curve was considered weak positive, no was negative.

**Table 2 vetsci-10-00689-t002:** Detection of PCV3 in different samples.

Sample Type	Serum	Semen	Oral Fluid	Preputial Fluid
Param	28	28	28	28
Positive number	1	0	18	13
Positive rate (%)	3.57	0	64.28	46.4

**Table 3 vetsci-10-00689-t003:** Distribution of PCV3 positive samples in different samples.

Type of Positive Sample	A	B	C	D	E
Number of Boars tested	28	28	28	28	28
Positive number	0	9	4	8	1
Percentage (%)	0	32.1	14.3	28.6	3.6

A: Only in serum, B: Only in oral fluid, C: Only in preputial fluid, D: Oral fluid and preputial fluid, E: Oral fluid and preputial fluid and serum.

## Data Availability

The original contributions presented in the study are included in the article, further inquiries can be directed to the corresponding author/s.

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
