# Peer review of "Detection of Porcine Circovirus Type 3 in Serum, Semen, Oral Fluid, and Preputial Fluid Samples of Boars"

_vetsci, 2023, doi:10.3390/vetsci10120689_

Round 1

Reviewer 1 Report

Comments and Suggestions for Authors

The Authors describe a study about the presence of porcine circovirus type 3 in different biological samples of boars, from a pig farm where PCV3 was already detected in colostrum as stillbirth and mummified fetuses. The research is quite well described and could give a good contribution to knowledge on this topic. However, some points, especially in Results and Discussion chapters, need revision or explanation by the Authors (often probably for a misunderstanding linked to english language), as suggested below.

Line 3 and following: in the title and overall in the text it’s used the term “urinary hydrocele” for which it’s not clear the meaning. Hydrocele is a well defined pathology, as known a scrotal swelling that occurs when fluid collects in the thin sheath that surrounds the testicle: are the boars suffering from hydrocele or in contrast what do the Authors intend to mean with this type of sample?

Line 38: I suggest to eliminate newly; as specified in lines 52 and following, PCV3 has been discovered already ten years ago (2013);

Line 55: PCV or PCV3?

Lines 87: specify what kind of ELISA, indirect or competitive;

Line 91 and 103: the titles are the same; I suggest to change them, for example “Animal source” (from which the samples have been collected) in line 91 and only “Samples” in line 103;

Line 179: as it’s topic for discussing the results about the presence of IgG in serum and oral fluids, declare which type of sample the kit has been made for by manufacturer;

Line 202: according to Table 1 and 2, PCV3 was not “only” detected in oral fluid and “urinary hydrocele” but also in one boar serum;

Lines 202-205: refer correctly lines 202-203 (until "hydrocele".) to Table 3 and lines 204-205 to Table 1 (in this last one it’s evident that 6 boars were negative for PCV3);

Line 203: "In one boar"…but PCV3 in both urinary hydrocele and oral fluid was detected in 9 boars according to Table 1. Check the results.

Line 208: explain what do the Authors mean with CT (also for better understanding following Table 1 and Discussion);

Line 208: the line “The CT value….35.5” repeats line 213 “Ct value was 35.5…” (I suggest to eliminate it);

Line 230: For obtaining a comparison of all results, I suggest strongly to insert the results of ELISA (with PP value) in the Table 1 for each boar;

Line 240: as PCV2, PPV, etc are virus and not diseases I suggest to change “diseases” in “virus”;

Line 251: It’s questionable that an higher incidence of PCV3 in fetuses indicate a transmission of PCV3 through boar semen (and obviously not a vertical transmission by mother to fetus); I suggest to reformulate the sentence;

Lines 260-261: it’s hypothesized that the “urinary hydrocele” (but as explained for line 2 it’s not clear what is this) could contaminate the semen; but if it’s real,  which are the possible reasons -if “urinary hydrocele” has been extruded just before the semen collection, so infecting the internal wall of uretra – for explaining the PCV3 negativity of semen in all boars?

Line 272-275: check the expression of these results as they are confused and somewhat in contrast with Table 1. For example, 1) if the line 272 is referred to oral fluid the lowest CT is 28.9 and not 28.1; 2) “only one sample had CT value exceeding 37” for what? 3) “Notably the CT value for urinary hydrocele was higher that that of oral fluid…”: I’ve not calculated the average of the values but Table 1 shows the opposite.

Line 293-295: as discussed for line 178, if the kit was intended only for serum samples it’s not possible draw conclusions about the presence of PCV3 in oral fluids stated in line 293-294. This aspect compromises the discussion drawed in line 303-306: I suggest to avoid any conclusion about the presence of IgG in oral fluids because, as it’said by the same Authors in line 294-295, the results are not reliable.

Lines 302-303: these aspects must first to be presented to the readers in the Results chapter before discuss them here. There is no comparative table where for example it’s possible to understand that only one boar showed negativity for both PCV3 PCR and serum IgG. Refers again to the comment for line 230.

Lines 308-310: these are information to be included in Material and Methods chapter.

Figure 1: I suggest to eliminate this Figure as it doesn’t add significative informations.

A final question/suggestion: many papers refers that porcine circoviruses spread in the body inside the monocytes: why the Author have not searched PCV3 by PCR in whole blood (or in buffy coat) as well as in serum?

Author Response

Dear reviewer:

Thank you for your letter and for the your comments concerning our manuscript entitled “Detection of porcine circovirus type 3 in serum, semen, oral fluid, and urinary hydrocele samples of boars” (Manuscript ID: vetsci-2692452). Those comments are all valuable and very helpful for revising and improving our paper, as well as the important guiding significance to our researches.

We have studied comments carefully and have made correction which we hope meet with approval. Revised portion are marked in red in the paper. The main corrections in the paper and the responds to the reviewer’s comments are as flowing:

Reviewer #1:

  1. Line 3 and following: in the title and overall in the text it’s used the term “urinary hydrocele” for which it’s not clear the meaning. Hydrocele is a well defined pathology, as known a scrotal swelling that occurs when fluid collects in the thin sheath that surrounds the testicle: are the boars suffering from hydrocele or in contrast what do the Authors intend to mean with this type of sample?

Response: We are very sorry for our incorrect writing. We found the correct terminology name “preputial fluid” on page 1236 of the 10th edition of Diseases of Swine, and we have made the replacement in the whole manuscript.

  1. Line 38: I suggest to eliminate newly; as specified in lines 52 and following, PCV3 has been discovered already ten years ago (2013).

Response: We are very sorry for our incorrect writing. We have re-written this part according to your suggestion.

  1. Line 55: PCV or PCV3?

Response: We are very sorry for our incorrect writing. I have modified it to “PCV3”.

4.Lines 87: specify what kind of ELISA, indirect or competitive.

Response: We are very sorry for our negligence. After our inspection confirmed, it’s indirect ELISA, and we have modified this imprecise expression.

  1. Line 91 and 103: the titles are the same; I suggest to change them, for example “Animal source” (from which the samples have been collected) in line 91 and only “Samples” in line 103.

Response: We have made correction according to your comments.

  1. Line 179: as it’s topic for discussing the results about the presence of IgG in serum and oral fluids, declare which type of sample the kit has been made for by manufacturer.

Response: It is really true as you pointed out the inaccurate description. We have confirmed that the serum is the only type of sample the kit has been made for by manufacturer, and we have deleted the description of PCV3-IgG antibody test data for oral fluid in the whole manuscript.

  1. Line 202: according to Table 1 and 2, PCV3 was not “only” detected in oral fluid and “urinary hydrocele” but also in one boar serum.

Response: We are very sorry for our easily misunderstood writing. We have re-written this part, which means the rate of PCV3 detection solely in oral fluid, excluding serum, semen, or preputial fluid samples, was found to be 32.1%. And we also modified the comments in Table 3.

  1. Lines 202-205: refer correctly lines 202-203 (until "hydrocele".) to Table 3 and lines 204-205 to Table 1 (in this last one it’s evident that 6 boars were negative for PCV3).

Response: We have re-written this part according to your suggestion.

  1. Line 203: "In one boar"…but PCV3 in both urinary hydrocele and oral fluid was detected in 9 boars according to Table 1. Check the results.

Response: We have re-written this part according to your suggestion.

  1. Line 208: explain what do the Authors mean with CT (also for better understanding following Table 1 and Discussion).

Response: Considering your suggestion, we have supplemented the mean of CT values from line 239 to 243.

  1. Line 208: the line “The CT value….35.5” repeats line 213 “Ct value was 35.5…” (I suggest to eliminate it).

Response: Thanks for your suggestion, and we have deleted it and improved the expression of the sentence.

  1. Line 230: For obtaining a comparison of all results, I suggest strongly to insert the results of ELISA (with PP value) in the Table 1 for each boar.

Response: Thanks for your suggestion, and we have inserted the results of ELISA (with PP value) in the Table 1 for each boar.

  1. Line 240: as PCV2, PPV, etc are virus and not diseases I suggest to change “diseases” in “virus”.

Response: Thanks for your suggestion, we have changed “diseases” in “virus”.

  1. Line 251: It’s questionable that a higher incidence of PCV3 in fetuses indicate a transmission of PCV3 through boar semen (and obviously not a vertical transmission by mother to fetus); I suggest to reformulate the sentence.

Response: We have re-written this part according to your suggestion. And we added some discussions about it from 321 to 338.

  1. Lines 260-261: it’s hypothesized that the “urinary hydrocele” (but as explained for line 2 it’s not clear what is this) could contaminate the semen; but if it’s real, which are the possible reasons -if “urinary hydrocele” has been extruded just before the semen collection, so infecting the internal wall of uretra – for explaining the PCV3 negativity of semen in all boars?

Response: Thanks for your suggestion, it is really true as you suggested, this pare is truly not a very scientific description, so we deleted this part.

  1. Line 272-275: check the expression of these results as they are confused and somewhat in contrast with Table 1. For example, 1) if the line 272 is referred to oral fluid the lowest CT is 28.9 and not 28.1; 2) “only one sample had CT value exceeding 37” for what? 3) “Notably the CT value for urinary hydrocele was higher that that of oral fluid…”: I’ve not calculated the average of the values but Table 1 shows the opposite.

Response: We are very sorry for our incorrect writing, we have re-written this part and made a more precise description

  1. Line 293-295: as discussed for line 178, if the kit was intended only for serum samples it’s not possible draw conclusions about the presence of PCV3 in oral fluids stated in line 293-294. This aspect compromises the discussion drawed in line 303-306: I suggest to avoid any conclusion about the presence of IgG in oral fluids because, as it’said by the same Authors in line 294-295, the results are not reliable.

Response: It is really true as you suggested, so we deleted this part.

  1. Lines 302-303: these aspects must first to be presented to the readers in the Results chapter before discuss them here. There is no comparative table where for example it’s possible to understand that only one boar showed negativity for both PCV3 PCR and serum IgG. Refers again to the comment for line 230.

Response: It is really true as you suggested, and we have re-written this part from line 266 to 275.

  1. Lines 308-310: these are information to be included in Material and Methods chapter.

Response: Thanks for your suggestion, it is really true as you suggested. We have transferred this part of the description to Material and Methods chapter.

  1. Figure 1: I suggest to eliminate this Figure as it doesn’t add significative informations”

Response: Thanks for your suggestion, we have eliminated Figure 1.

  1. A final question/suggestion: many papers refers that porcine circoviruses spread in the body inside the monocytes: why the Author have not searched PCV3 by PCR in whole blood (or in buffy coat) as well as in serum?.

Response: Thanks for your professional suggestion. I have reviewed some relevant literatures, just like your suggestion, it is indeed accurate, and we have made some discussions based on your suggestions from line 347 to 354.

Special thanks to you for your good comments.

We tried our best to improve the manuscript and made some changes in the manuscript. These changes will not influence the content and framework of the paper. And here we did not list the changes but marked in red in revised paper.

We appreciate for your warm work earnestly, and hope that the correction will meet with approval.

Once again, thank you very much for your comments and suggestions.

Reviewer 2 Report

Comments and Suggestions for Authors

The study involved the examination and analysis of fresh semen, as well as matched serum, oral fluid, and urinary hydrocele samples collected from 28 boars on a large-scale pig farm located in Guangxi, China. The findings suggest that PCV3 may have limited ability to transmit through semen, which has important implications for the development of targeted interventions to reduce the risk of transmission through artificial insemination. Overall, I found the study to be well-conducted and the results to be informative. However, I have several specific points that I believe could improve the manuscript:

1. The study's sample size is limited to 28 artificially inseminated boars from a single pig farm in Guangxi Province. While the results are interesting, it is unclear how generalizable they are to other pig populations. I suggest discussing the limitations of the sample size and how the findings might be applied to other populations.

2. The manuscript would benefit from a more detailed description of the methods used to detect PCV3 in the various samples. Specifically, it would be helpful to know more about the sensitivity and specificity of the PCV3 fluorescence quantitative Polymerase Chain Reaction (qPCR) kit and the PCV3-IgG ELISA kit.

3. The study's findings suggest that PCV3 may have limited ability to transmit through semen. However, the authors do not discuss the potential implications of this finding for the development of targeted interventions to reduce the risk of transmission through artificial insemination. I suggest expanding on this point in the discussion section.

4. Finally, the manuscript would benefit from a more detailed discussion of the potential implications of the study's findings for the control and prevention of PCV3 in pig populations. Specifically, it would be helpful to know more about how the findings might be applied to the development of targeted interventions to reduce the risk of transmission through direct oral-nasal contact.

Comments on the Quality of English Language

some English grammar issues have been observed.

Author Response

Dear reviewers:

Thank you for your letter and for your comments concerning our manuscript entitled “Detection of porcine circovirus type 3 in serum, semen, oral fluid, and urinary hydrocele samples of boars” (Manuscript ID: vetsci-2692452). Those comments are all valuable and very helpful for revising and improving our paper, as well as the important guiding significance to our researches.

We have studied comments carefully and have made correction which we hope meet with approval. Revised portion are marked in red in the paper. The main corrections in the paper and the responds to the reviewer’s comments are as flowing:

Reviewer #2:

  1. The study's sample size is limited to 28 artificially inseminated boars from a single pig farm in Guangxi Province. While the results are interesting, it is unclear how generalizable they are to other pig populations. I suggest discussing the limitations of the sample size and how the findings might be applied to other populations.

Response: Thanks for your suggestion, we have made some discussions about this point from line 442 to 451.

  1. The manuscript would benefit from a more detailed description of the methods used to detect PCV3 in the various samples. Specifically, it would be helpful to know more about the sensitivity and specificity of the PCV3 fluorescence quantitative Polymerase Chain Reaction (qPCR) kit and the PCV3-IgG ELISA kit.

Response: It is really true as you suggested. Regarding the sensitivity and specificity of the PCV3 qPCR kit and PCV3-IgG ELISA kit, we have cited the original sources in the paper, and these methods have been well validated in two literatures, which clarified and explained the detection sensitivity and specificity of the kit. From line 101 to 104 and 285 to 288.

  1. The study's findings suggest that PCV3 may have limited ability to transmit through semen. However, the authors do not discuss the potential implications of this finding for the development of targeted interventions to reduce the risk of transmission through artificial insemination. I suggest expanding on this point in the discussion section.

Response: Considering your suggestion, we have added some discussions about how to reduce the risk of transmission through AI, and quoted some literatures, from 321 to 340.

  1. Finally, the manuscript would benefit from a more detailed discussion of the potential implications of the study's findings for the control and prevention of PCV3 in pig populations. Specifically, it would be helpful to know more about how the findings might be applied to the development of targeted interventions to reduce the risk of transmission through direct oral-nasal contact.

Response: Considering your suggestion, we have added some discussions about how to reduce the risk of transmission through direct oral-nasal contact, from 372 to 381.

Special thanks to you for your good comments.

We tried our best to improve the manuscript and made some changes in the manuscript. These changes will not influence the content and framework of the paper. And here we did not list the changes but marked in red in revised paper.

We appreciate for your warm work earnestly, and hope that the correction will meet with approval.

Once again, thank you very much for your comments and suggestions.

Reviewer 3 Report

Comments and Suggestions for Authors

This manuscript explores the detection of porcine circovirus type 3 in serum, semen, oral fluid, and urinary hydrocele samples of boars. Prior to publication, several points necessitate clarification and refinement, as outlined below:

1.

Definition of Positive Detection Rate and False Positives:

The manuscript should provide a clearer definition of the positive detection rate. Additionally, clarification is needed regarding the criteria used as the gold standard for determining false positives.

2.

Abstract Clarification:

In the abstract, the authors assert that "The results of this study preliminary clarify that PCV3 is transmitted among boars on pig farms and exhibits epidemic characteristics." However, it is crucial to note that the samples were exclusively obtained from 28 artificially inseminated boars, aged 18 to 35 months, sourced from a specific pig farm in Guangxi Province. Consequently, the results may only infer transmission patterns within this particular farm.

3.

Sensitivity of Detection Methods:

The manuscript should elucidate the sensitivity of the commercially available PCV3-IgG ELISA kit used for PCV3-IgG antibody detection in serum and oral fluid samples. Furthermore, a comparative analysis of the sensitivity of this ELISA kit in relation to the qPCR method, employed for detecting porcine circovirus type 3 in various samples, is needed.

4.

Mechanism of Non-Detection in Semen:

The statement "These findings suggest that PCV3 may have limited ability to penetrate the blood-testosterone barrier and transmit through semen" requires further elaboration. Specifically, the authors should elucidate the mechanism underlying the non-detection of PCV3 in semen. Is this attributed to the absence of PCV3 in semen, or does it result from limitations in DNA extraction and detection methods?

5.

Sample Size Consideration:

A concern is raised regarding the limited number of samples analyzed. Authors are encouraged to address this issue by providing a rationale for the chosen sample size and discussing the potential implications of the sample size on the study's findings and generalizability.

Comments on the Quality of English Language

The manuscript demonstrates a commendable quality of English language

Author Response

Dear reviewers:

Thank you for your letter and for your comments concerning our manuscript entitled “Detection of porcine circovirus type 3 in serum, semen, oral fluid, and urinary hydrocele samples of boars” (Manuscript ID: vetsci-2692452). Those comments are all valuable and very helpful for revising and improving our paper, as well as the important guiding significance to our researches.

We have studied comments carefully and have made correction which we hope meet with approval. Revised portion are marked in red in the paper. The main corrections in the paper and the responds to the reviewer’s comments are as flowing:

Reviewer #3:

  1. Definition of Positive Detection Rate and False Positives: The manuscript should provide a clearer definition of the positive detection rate. Additionally, clarification is needed regarding the criteria used as the gold standard for determining false positives.

Response: It is really true as you suggested. Regarding the sensitivity and specificity of the PCV3 qPCR kit and PCV3-IgG ELISA kit, we have cited the original sources in the paper, and these methods have been well validated in two literatures, which clarified and explained the detection sensitivity and specificity of the kit. From line 101 to 104 and 285 to 288.

  1. Abstract Clarification:

In the abstract, the authors assert that "The results of this study preliminary clarify that PCV3 is transmitted among boars on pig farms and exhibits epidemic characteristics." However, it is crucial to note that the samples were exclusively obtained from 28 artificially inseminated boars, aged 18 to 35 months, sourced from a specific pig farm in Guangxi Province. Consequently, the results may only infer transmission patterns within this particular farm.

Response: Thanks for your suggestion, we have made some discussions about this point from line 442 to 451.

  1. Sensitivity of Detection Methods:

The manuscript should elucidate the sensitivity of the commercially available PCV3-IgG ELISA kit used for PCV3-IgG antibody detection in serum and oral fluid samples. Furthermore, a comparative analysis of the sensitivity of this ELISA kit in relation to the qPCR method, employed for detecting porcine circovirus type 3 in various samples, is needed.

Response: It is really true as you suggested. Regarding the sensitivity and specificity of the PCV3 qPCR kit and PCV3-IgG ELISA kit, we have cited the original sources in the paper, and these methods have been well validated in two literatures, which clarified and explained the detection sensitivity and specificity of the kit. From line 101 to 104 and 285 to 288.

  1. Mechanism of Non-Detection in Semen:

The statement "These findings suggest that PCV3 may have limited ability to penetrate the blood-testosterone barrier and transmit through semen" requires further elaboration. Specifically, the authors should elucidate the mechanism underlying the non-detection of PCV3 in semen. Is this attributed to the absence of PCV3 in semen, or does it result from limitations in DNA extraction and detection methods?

Response: Thanks for your professional suggestions, and according to your suggestion, we have made some discussions in this part by citing relevant literature, from line 340 to 351.

  1. Sample Size Consideration:

A concern is raised regarding the limited number of samples analyzed. Authors are encouraged to address this issue by providing a rationale for the chosen sample size and discussing the potential implications of the sample size on the study's findings and generalizability.

Response: Thanks for your suggestion, we have cited relevant references and made some discussions about this point from line 442 to 451.

We tried our best to improve the manuscript and made some changes in the manuscript. These changes will not influence the content and framework of the paper. And here we did not list the changes but marked in red in revised paper.

We appreciate for your warm work earnestly, and hope that the correction will meet with approval.

Once again, thank you very much for your comments and suggestions.

Round 2

Reviewer 1 Report

Comments and Suggestions for Authors

Thanks to the Author for accepting all my suggestions and correcting accordingly the paper.

I indicate the points that for me still need a revision.

Line 160: correct “Animals sources were obtained…” in “Samples were obtained….” (the 28 boars are themselves “animal sources”). A note (I suggest strongly to check with attention the english language through all the paper for avoiding frequent confusions): which is the meaning of “artificially inseminated boars”? Boars born from an artificially inseminated sow? Or probably boars from which the semen is collected for artificial insemination of sows (in other words, semen donors)?

 Line 243: in order to standardize the definition of CT in line 248 and line 311 (and everywhere in the text if necessary), use this last one: Cycle Threshold (CT).

Line 168: as yet suggested, correct in “those other virus associated…” as PRRSV, PPV etc are virus and not diseases.

Line 414 and 433: as yet suggested in the first revision, pay attention to the use of “vertical transmission” speaking about semen; infected semen or preputial fluid as all other body fluids would be eventually responsible of “horizontal transmission” considering - as commonly accepted - “vertical transmission” only that from mother to fetus through the placenta.

Lines 532-535: I suggest to eliminate or changing these lines as confused. The (detection) sensitivity is a quality of analytical test (as the specificity) and it’s not referred to a capacity or a probability of a particular sample to host a pathogen (in this case preputial fluid); “this approach ensures preservation of sensitivity and sensitivity”, probably the last one specificity, should be proven relating true and false infected animals to the results of the tests used, what was not done in the present work.

Author Response

Dear Reviewer

On behalf of my co-authors, we thank you very much again for giving us an opportunity to revise our manuscript, and we also appreciate reviewers very much for their positive and constructive comments and suggestions on our manuscript entitled “Detection of porcine circovirus type 3 in serum, semen, oral fluid, and preputial fluid samples of boars” (Manuscript Number: vetsci-2692452).

We revised the manuscript according to these comments and suggestions. In general, we have tried our best to revise our manuscript and provide the point-by-point responses. All changes were marked in red (blue was the modification, and maybe because of the file format, the line number you marked is different from mine) using the “Track Changes” function in the revised manuscript. Attached please find our responses to your kindness comments.

The following is a summary list of changes: 

Comment 1:

Line 160: correct “Animals sources were obtained…” in “Samples were obtained….” (the 28 boars are themselves “animal sources”). A note (I suggest strongly to check with attention the English language through all the paper for avoiding frequent confusions): which is the meaning of “artificially inseminated boars”? Boars born from an artificially inseminated sow? Or probably boars from which the semen is collected for artificial insemination of sows (in other words, semen donors)?

Response: Thanks for your professional comment, we have eliminate “artificially inseminated”, line 127.

Comment 2:

 Line 243: in order to standardize the definition of CT in line 248 and line 311 (and everywhere in the text if necessary), use this last one: Cycle Threshold (CT).

Response: Thanks for your professional comment, we have changed this sentence to “A cut-off for positive samples was established at cycle threshold (CT) values lower than 40” in line 194. And in line 257 we deleted the repetition writing.

Comment 3:

Line 168: as yet suggested, correct in “those other virus associated…” as PRRSV, PPV etc are virus and not diseases.

Response: We are very sorry that we have ignored the modification here as your suggestion, and we have changed “diseases” to “viruses”, in line 136.

Comment 3:

Line 414 and 433: as yet suggested in the first revision, pay attention to the use of “vertical transmission” speaking about semen; infected semen or preputial fluid as all other body fluids would be eventually responsible of “horizontal transmission” considering - as commonly accepted - “vertical transmission” only that from mother to fetus through the placenta.

Response: Thanks for your professional comment, we have eliminate “vertical transmission”, in line 343 and 348

Comment 5:

Lines 532-535: I suggest to eliminate or changing these lines as confused. The (detection) sensitivity is a quality of analytical test (as the specificity) and it’s not referred to a capacity or a probability of a particular sample to host a pathogen (in this case preputial fluid); “this approach ensures preservation of sensitivity and sensitivity”, probably the last one specificity, should be proven relating true and false infected animals to the results of the tests used, what was not done in the present work.

Response: Your professional comment is very correct, and we have learned this point, and we have eliminated these lines as your suggestion. From line 406-410.

Once again, thank you very much for your comments and suggestions.

Thank you and best regards.

Sincerely yours

Reviewer 3 Report

Comments and Suggestions for Authors

Accept

Comments on the Quality of English Language

Good

Author Response

(The authors gave the same response as above.)
